# Reproducibility Report:
# Towards Visually Explaining Variational Autoencoders

## Reproducibility Summary

**Scope of Reproducibility**

Using a modification of Grad-CAM, attention maps can be created for Variational Autoencoders, resulting in explainable generations. Using these attention maps, state-of-the-art anomaly detection and latent space disentanglement is reached.

**Methodology**

We started the challenge using the author's code, but this only covered one experiment of the paper, namely anomaly detection for the MNIST dataset. Therefore we added models, training and testing code for all other anomaly detection experiments, those on the UCSD-Ped1 and MVTec dataset, and also for the latent space disentanglement experiments. Some of these implementations were based on other existing repositories, whereas some were implemented completely by ourselves. We worked for four weeks full-time on reproducing the results with two GPUs available to use.

**Results**

We were able to successfully generate attention maps using the method described by Liu et al. and could apply them to anomaly detection as well. For the MNIST experiments, this led to results that were similar to the paper. However, for the UCSD-Ped1 experiments, the author's explainable VAE model actually performed worse than our own baseline. Moreover, we were not able to support the author's claim that they achieve state-of-the-art on the MVTec dataset. Finally, for the latent space disentanglement, our found results were not as good as claimed by Liu et al., but they still out-performed the set baseline, as was also claimed by the authors.

**What was easy**

Running the initial implementation of the authors, since their code was relatively straightforward. We were able to generate attention maps and anomaly detections for the MNIST dataset using a Variational Autoencoder without too many difficulties.

**What was difficult**

The code of the authors covered only a small portion of the paper and extending this to the whole paper was very difficult, as the paper was not often very clear on the implementation details. Adding in certain metrics for evaluation turned out to be relatively hard as well.

**Communication with original authors**

We contacted the authors by email, as provided in their paper and on Github, but were not answered. Another group within our course working on the same paper did get a response, that way we got some additional insights.

Submitted to ML Reproducibility Challenge 2020. Do not distribute.

# 1  Introduction

Recently there has been an increasing interest in model explainability within artificial intelligence research. One branch of study concerns itself with generating visual explanations, or visual attention. These visual explanations highlight the areas in images or other visual data that the model deems most important in making correct predictions, thus essentially explaining how the model reasons. This makes the inner workings of AI models more transparent. Visual attention techniques have so far mainly been applied to Convolutional Neural Networks (CNN), to visualize the regions of an image that are most important for making a classification. However, visual explanations have not yet been applied to many generative models.

Liu et al. attempt to bridge this gap with their paper "Towards Visually Explaining Variational Autoencoders". They describe a technique for generating visual attention for Variational Autoencoders (VAE) [Kingma and Welling, 2014], a type of generative model. With their visual attention method for VAEs, Liu et al. take a step towards making AI models more transparent. We find this to be important, because as AI grows more prominent, so does the desire for the models to be explainable. If AI is to take a central spot in our lives and society, then the decisions it makes need to be transparent. This makes the models safer, less subjective to manipulation and makes people more likely to trust the decisions made by the model. As generative models have mostly been black-box type up until now, the claims made in the paper by Liu et al. are very promising for the field of transparent AI. We will therefore attempt to reproduce this paper, giving more insight in the validity of the made claims.

The three major claims are made in the paper are as follows:

- It is possible to generate visual attention maps conditioned on the latent space of a VAE using a method based on Grad-CAM.
- Using these attention maps, it becomes possible to achieve state-of-the-art performance for an anomaly localization task on the MVTec-AD dataset.
- The attention maps can also be incorporated into a new learning objective called attention disentanglement loss, which improves upon the state-of-the-art in latent space disentanglement for VAEs.

In this report, we will attempt to validate all three claims by reproducing the described experiments. The choice to reproduce all three is made because the first claim contains the core implementation and the second and third claim state-of-the-art results. Thus by evaluating all three claims we aim to ascertain the strengths and weaknesses of the paper and underlying code. Our results show that the first claim can be reproduced, as we got results that were similar to the paper. As for the two other claims, we were able to implement all experiments, but were not able to reach the same state-of-the-art results as the authors. This was the case for both the MVTec-AD dataset and the latent space disentanglement experiments, therefore leading us to not being able to fully support the second and third claim.

# 2  Methodology

## 2.1  Concepts and Models

The below subsections will further elaborate on the theory behind each of the three claims and the model implementations we used to run the corresponding experiments.

### 2.1.1  VAE Attention

The first claim the paper makes is that they can generate and visualize attention for VAEs, which was previously only applicable to CNNs. The method they describe for this is based on a technique called Grad-CAM, which stands for Gradient-weighted Class Activation Mapping [Selvaraju et al., 2017]. Grad-CAM uses gradients for visualizing the regions in an image that are most important for classification. It does this by computing gradients backpropagated from the classifier unit to a target convolutional layer, thus generating a feature map $\mathbf{M}$. In Liu et al., this method is extended to work on VAEs. The key difference is that, for the method proposed by Liu et al., the gradients are not backpropagated from a CNN's classification unit, but from a latent vector $z$ of the VAE. For each $z_i$ in $\mathbf{z}$, the corresponding attention map $\mathbf{M}^i$ is computed by backpropagating gradients to the target layer feature maps $\mathbf{A} \in \mathbb{R}^{n \times h \times w}$. Details of this method can be found in equations 2 and 3 in the original paper by Liu et al..

Via the website of Papers with Code[1], we found a GitHub repository from one of the authors that corresponded with the paper[2]. However, the majority of the required code to reproduce all three claims was absent. For example, there was no

---

[1] https://paperswithcode.com/paper/towards-visually-explaining-variational
[2] https://github.com/liuem607/expVAE

code available for the particular attention generation method described above. Therefore, we wrote the implementation ourselves. This allowed us to access the individual feature maps $\mathbf{M}^i$ for each $z_i$, which were also later needed for the implementation of the AD loss function.

### 2.1.2 Anomaly Detection

The second claim the authors make is that they can reach state-of-the-art performance for anomaly detection using their VAE attention method. To implement the attention generating mechanism for anomaly detection, a slightly different method than the one described above was used. Instead of computing individual attention maps $\mathbf{M}^i$ for each element in the latent vector $\mathbf{z}$, they take the inferred mean vector of the latent space and sum it to compute the score $s$, which is then backpropagated to the target layer.

The GitHub repository of the paper contained an implementation of the aforementioned method. This code however did not include any documentation or comments and there were also blocks of code present that were never called. We removed these parts from the code, slightly restructured it to be more efficient and added documentation to each of the implemented functions. Another notable feature in the authors their code, was the absence of a ReLU function for the attention map, as was originally mentioned in the paper. In its place was an absolute operation that serves as an alternative to the ReLU to eliminate negative values for the attention maps. However, we think the ReLU only makes sense for the original Grad-CAM paper Selvaraju et al. [2017], and that only the magnitude of the attention and not the sign is important for the VAE. After this inspection, we also moved the absolute operation inside the summation of the attention maps, which improved our results.

Two different models were implemented for the anomaly detection task. The first model the authors implement, called Vanilla expVAE, applies the attention generation technique described above to a relatively simple one-class VAE. The exact details of this model's architecture were not mentioned in the paper. However, via the FACT-AI course where this report is part of, we received a supplementary document originally created by the authors which gave more details into the exact architectures and some hyperparameters. We used the same architecture for the Vanilla expVAE as described in the document, which can be found in Appendix A. The second model implemented by the paper applies the VAE attention mechanism to a VAE model with Resnet18 CNN architecture as an encoder. A precise implementation of the Resnet18 expVAE as described in the supplemented materials turned out to be unattainable since the authors mention a $512 \times 8 \times 8$ output size at two layers before the output, however the Resnet18 output has a size of $512 \times 16 \times 16$ at this point. To this end, we instead used the original Resnet18 implementation as the encoder but kept the decoder the same as the authors described. Exact details of this architecture can also be found in appendix A.

### 2.1.3 Attention Disentanglement

The third claim made by the authors is that they can reach state-of-the-art performance in latent space disentanglement by incorporating their attention maps in an already existing disentanglement model called FactorVAE [Kim and Mnih, 2018]. The original FactorVAE improved upon the $\beta$-VAE [Higgins et al., 2016] by overcoming the trade-off between disentanglement and reconstruction quality inherent to the $\beta$-VAE, reaching state-of-the-art performance on latent space disentanglement whilst not hurting the reconstructions. The assumption is that, if a VAE is completely disentangled, one latent dimension will correspond to one latent factor in the data and when a latent traversal is done over this latent dimension, the transformation will be similar to traversing the corresponding latent factor.

Liu et al. claim that they can improve on this by adding an additional loss module based on the generated attention maps, which they call Attention Disentanglement (AD) loss. The AD loss uses two attention maps computed from different latent dimensions and increases based on the overlap between both maps. This corresponds to equation 5 in [Liu et al.]. Note that the $\mathbf{A}$ for this equation does not represent the same thing as the one in equation 2 of their paper.

As Liu et al. use the original FactorVAE architecture, our code builds upon an open-source implementation by WonKwang Lee[3]. This implementation missed two parts which are required for reproducing the results: (1) the disentanglement metric proposed by Kim and Mnih was not implemented and (2) the AD-loss and thus also a version of Grad-CAM for generating attention maps per latent dimension needed to be implemented.

For (1), the disentanglement metric is based on a majority vote classifier. A vote corresponds to which factor, a variable attribute of the dataset, is aligned with which latent dimension of the VAE, see [Kim and Mnih, 2018] for a more detailed explanation. An almost complete implementation of the metric was found online[4]. It was slightly modified, mainly in terms of efficiency, and ported to PyTorch from TensorFlow. Note that we, following the example of this

---

[3]The GitHub repository can be found here `https://github.com/1Konny/FactorVAE`

[4]See the function *evaluate_disentanglement* in `https://github.com/nicolasigor/FactorVAE/blob/master/vae_dsprites_v2.py`.

implementation, only use the mean outputs of the encoder to retrieve the variance instead of reparameterizing the encoder outputs, which allowed for the baseline FactorVAE to get similar results as achieved by Liu et al.. This is because using the means will create a more stable representation and thus also a more stable/higher score, whilst still being faithful to the output latents, as these will be centred around the means.

For (2), the modified implementation of Grad-CAM by Liu et al. was used to create attention maps for two latent dimensions. As mentioned in section 2.1.1, for this purpose too, alterations had to be made to the original code to allow for the creation of attention maps of individual latent dimensions. The ReLU activation (from equation 2 of [Liu et al.]) was still used here, as the focus here is on positive attention (the loss should not be negative). In the paper it is never exactly stated how the two latent dimensions for the attention maps were chosen, we chose those randomly at each iteration during training. Retrieving the attention map for each latent dimension to allow for a different implementation seemed unwise: calculating the loss for two maps already resulted in an efficiency drop of around 35 per cent (from 70 iterations per second to 45). This drop is likely due to the creation of the attention maps being computationally expensive, as it requires an extra backward step through the encoder network per latent dimension.

## 2.2 Datasets and hyperparameters

Using these models, Liu et al. ran experiments on multiple datasets: anomaly detection experiments were run for the MNIST, UCSD-Ped1 and MVTec datasets and latent space disentanglement on the dSprites dataset. This section gives an overview of these datasets, as well as the hyperparameters we used for reproducing the experiments during training and testing.

### 2.2.1 MNIST

The MNIST dataset is a dataset containing 60,000 training and 10,000 testing samples of black-and-white images of handwritten digits, ranging from zero to nine [LeCun et al., 2010]. A dataloader script, which downloads the required data and divides it into training and testing sets, was present in the author's GitHub repository. Therefore no further additions were required to process this dataset.

In the supplementary document provided by the authors, it was mentioned that the images were resized to $28 \times 28$ pixels. Furthermore, the learning rate was set to 0.001, the latent size to 32 and the batch size to 128. As for the number of epochs, no information was provided in the paper or the supplementary document. Therefore, we trained the network for a number of different epochs and determined from the resulting loss graphs that performance did not improve after 100 epochs anymore, so we used that as our number of training epochs. Moreover, the code by the author's used the Adam optimizer [Kingma and Ba, 2014] and a Binary Cross-Entropy (BCE) loss module, which made us decide to also use this for our experiments.

### 2.2.2 UCSD-Ped1

The USCD-Ped1 Anomaly Detection Dataset [Chan and Vasconcelos, 2008] is an open-source dataset containing 34 training and 36 testing samples of videos of a pedestrian walkway. Each video consists of 200 frames. In the training videos, pedestrians can be seen walking towards and away from the stationary camera overlooking the walkway. However, the testing videos also contain some anomalies such as bikers, skaters, small carts or pedestrians walking off the walkway. These anomalies are indicated using a mask. To train the pedestrian dataset all images were, like in the original paper, resized to $100 \times 100$ pixels and the latent size was set to 32. After consultancy with [Liu et al.] we used the VAE architecture as provided by the in appendix A figure 5 and a batch size of 32. We decided to train this architecture for 512 epochs and used BCE loss and the Adam optimizer. To evaluate this dataset, the Area Under the Receiver Operating Characteristic Curve (AUROC) was computed. In order to do so, the scikit-learn[Pedregosa et al., 2011] library was used. In addition, we recreated the baseline provided by the paper by computing the differences between the input images and their reconstructions. At last, we tried a new baseline by computing the difference between the average VAE reconstruction and the input image.

### 2.2.3 MVTec AD

MVTec AD [Bergmann et al., 2019] is a dataset that contains over 3929 training and 1725 testing samples of high-resolution images, divided into fifteen different object and texture categories. Specifically, these classes include bottles, cables, capsules, carpets, grids, hazelnuts, leather, pills, screws, tiles, toothbrushes, transistors, wood and zippers. For each of these classes, the training set contains anomaly-free image samples and the test set contains anomaly-free and anomalous images. Each class contains a variety of defects in the anomalous images, that are common for that object. On average 5 different types of defects per object. For all the anomaly images a binary mask image is present which marks the defected area of the object.

For the MVTec AD experiments, we again based most of our hyperparameters on the supplementary document. First of all, all images were resized to $256 \times 256$ pixels. Moreover, we applied data augmentation during training with random rotations between [-30, +30] degrees and random mirroring by horizontal and vertical flipping with a probability of 0.5. We set the learning rate 0.0001 and batch size to 8 for training and used a latent size of 32. Additional input processing was implemented by normalizing the input images by the mean and standard deviation of the entire dataset. This normalization technique is known to be a common approach when working with image data and is applied in many scientific papers to improve the performance of the network. Finally, the Adam optimizer was used and both Mean Squared Error (MSE) loss and BCE loss were compared. As BCE gave a slightly higher performance, we decided to use this as the loss module.

### 2.2.4 AD-FactorVAE

The dSprites dataset [Matthey et al., 2017] is a dataset developed by Deepmind to enable finding correlations between latent factors of the dataset and the dimensions of the latent dimensions of the VAE. The dataset is in a black and white colour space and consists of six latent factors: colour, shape (square, ellipse, heart), scale, orientation, x position and y position. Each factor has multiple classes (except for colour, which can only be white) which can be changed without influencing the other factors.

Liu et al. mention they build upon the original FactorVAE, of which the hyperparameters and architectures can be found in Appendix A of Kim and Mnih [2018]. We use the same hyperparameters and model architectures as Kim and Mnih as they are more efficient, but with latent dimensionality of 32, as used by Liu et al. (see figure 7), which allowed for the baseline FactorVAE to reach similar results as theirs. For adding the AD loss, a $\lambda$ term is mentioned, but it is never specified which value the authors use for this hyperparameter, thus we run experiments with $\lambda$ set at 1, 20, 40 and 80. In addition, the AD loss can be calculated for different convolutional layers in the FactorVAE, but the authors never explicitly mention which one is used. Therefor, we perform all experiments using the first convolutional layer and one extra experiment using the third convolutional layer with $\lambda = 1$.

### 2.3 Setup and Computational Requirements

The code for our experiments is made publicly available on GitHub[5], which includes python files and a notebook file to run all the experiments. The major part of the code was run using one GPU of the Lisa Cluster from SURFSARA (GeForce 1080Ti) [6], however the last part of our experiments concerning the attention disentanglement were run on our own GeForce 1080 GPU.

Using this hardware, one of the major computational costs arose from training the VAE on the MVTec AD dataset. On the Lisa Cluster, this took us around 45 minutes each, so for five classes, this meant up to four hours for 200 iterations. Furthermore, training the USCD-Ped1 dataset could take up to 4 hours for 512 iterations and training the FactorVAE took around 1 hour and 15 minutes per run (300000 iterations). Additionally, when adding the attention disentanglement loss, it took around 1 hour and 50 minutes per run, without the disentanglement metric being used. Adding this metric resulted in approximately twenty extra minutes per run.

## 3 Results

In order to validate the three claims made by Liu et al., we reproduced the experiments described in the paper. In their paper, after Liu et al. introduce their VAE attention mechanism, they apply the technique to anomaly detection, therefore showing that they are in fact able to produce VAE attention maps. For this reason, we have followed the same format and reproduced all of their anomaly detection experiments, thereby evaluating the validity of both claims 1 and 2. We then proceeded to evaluate the validity of their third claim, by also reproducing their latent space disentanglement experiments. The results of all of these experiments are discussed below.

### 3.1 Anomaly Detection

### 3.1.1 Evaluation on MNIST Dataset

In their second claim, Liu et al. state that their attention generation technique can be used for anomaly detection and is even able to achieve state-of-the-art on the MVTec AD dataset. The first anomaly detection experiment they describe is

---

[5]https://github.com/FrankBrongers/Reproducing_expVAE
[6]https://userinfo.surfsara.nl/systems/lisa/description

| | Liu et al. | Our reproduction | Our best |
|---|---|---|---|
| Baseline | 0.86 | 0.701 | **0.921** |
| Conv 1 | **0.89** | 0.468 | 0.552 |
| Conv 2 | **0.92** | 0.644 | 0.320 |
| Conv 3 | **0.91** | 0.802 | 0.858 |

(a) Quantitative results for the USCD-ped1 dataset compared to Liu et al. and its baseline. The "Our reproduction" column shows our results when using the authors model architecture. The "Our best" column shows the results of our model containing the highest scoring layer after hyperparameter search. The baseline score for "Our best" is our suggested new baseline by computing the difference between the average VAE reconstruction and the input image.

| Category | Liu et al. | Ours | Layer |
|---|---|---|---|
| Leather | **0.95** | 0.86 | layer2.0.conv2 |
| | 0.24 | 0.24 | |
| Tile | **0.80** | 0.73 | layer4.1.conv1 |
| | 0.23 | 0.17 | |
| Capsule | 0.74 | **0.90** | layer3.1.conv2 |
| | 0.11 | 0.07 | |
| Hazelnut | **0.98** | 0.93 | layer2.1.conv1 |
| | 0.44 | 0.26 | |
| Metal Nut | **0.94** | 0.67 | layer2.0.conv1 |
| | 0.49 | 0.18 | |

(b) Results for 5 categories from MVTec-AD dataset. For each category, we report the AUROC score on the top row, and best IOU on the bottom row.

Table 1: Pixel segmentation results for anomaly detection compared to the original paper. We adopt scores from Liu et al. for comparison.

a qualitative evaluation on the MNIST dataset, in which they train their model on the digit "1" and test on a variety of other digits. Below figure shows a selection of results for our reproduction of the experiment.

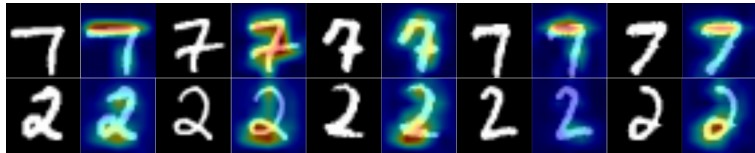

Figure 1: Resulting anomaly attention maps from training on "1" and testing on "7" and "2".

As can be seen in figure 1, there is some variation in the performance of the model. Some of the anomaly detection maps correctly highlight all parts of the digit that are anomalous to a "1". These results look very similar to the results Liu et al. shown in their paper in Figure 4. However, there are also some results where the whole digit, or no areas in the digit at all are highlighted.

### 3.1.2 Evaluation on USCD-Ped1 Dataset

Figure 2 shows qualitative anomaly detection results for different methods (c, d, e) compared to the input image and ground truth mask (a, b). Qualitative results are shown in table 1a, where our reproduction and best model are compared to the results of Liu et al.. Our results are substantially lower. Even though we only care about the highest performing layer, it is visible that the results are less consistent per layer. Even the obtained baseline score is lower. After adding batch normalization, a learning rate scheduler and using convolutional layers of depth [192, 144, 96] we achieve a minor increase in the obtained AUROC score as shown in the column "Our best". Note that our new suggested baseline score is as high as the best layer of Liu et al..

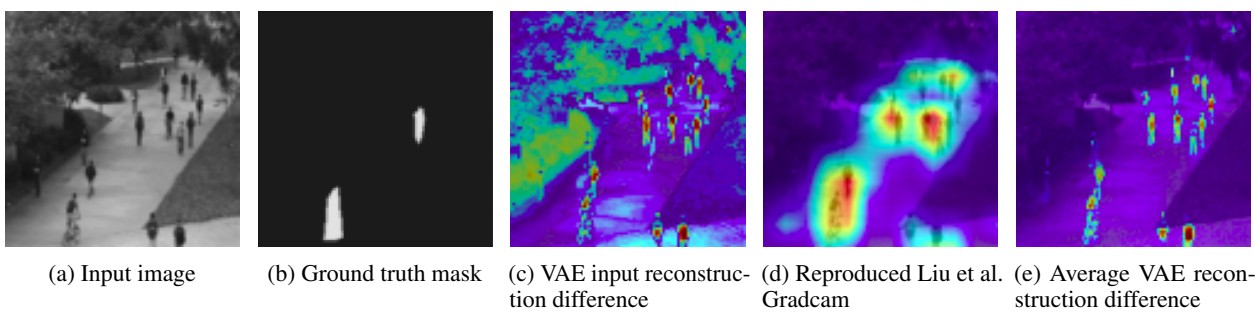

(a) Input image    (b) Ground truth mask    (c) VAE input reconstruction difference    (d) Reproduced Liu et al. Gradcam    (e) Average VAE reconstruction difference

Figure 2: Anomaly detection approaches (c, d, e) compared to the input image and ground truth mask (a, b).

### 3.1.3 Evaluation on MVTec AD

To evaluate the second claim, the author's results were reproduced for five different object classes: two of their best texture classes (leather and tile), two of their best object classes (hazelnut and metal nut) and their worst-performing class (capsule). An example of a generated attention map for the hazelnut class is shown in Appendix B. Table 1b shows the AUROC score and the best Intersection over Union (IoU). In addition, to make the results in , for transparency, we included the target layer that was used to generate the best score for each class. The AUROC scores show comparable results for hazelnut and leather, lacking results for metal nut, but much higher AUROC score for the capsule.

### 3.2 Latent Space Disentanglement

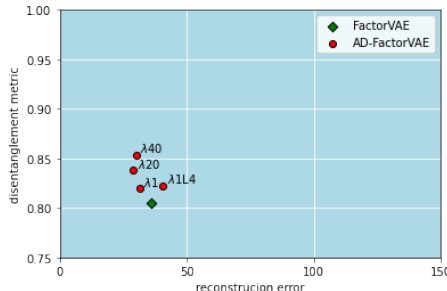

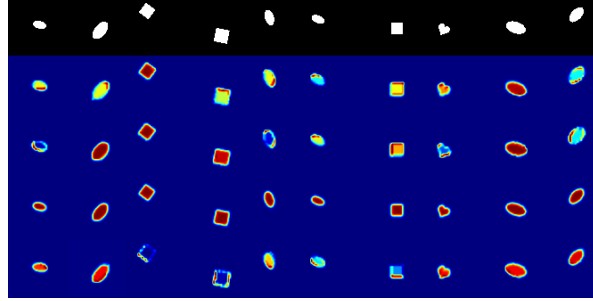

(a) Quantitative results of the FactorVAE against the AD-FactorVAE. All models are run with $\gamma = 40$ and averaged over three seeds, the number after the $\lambda$ indicates its corresponding value for that model and the number after L indicates the target layer, if this is not specified the target layer is the first layer. Note that the result for $\lambda = 80$ is similar to $\lambda = 1$ and was thus left out for clarity.

(b) Qualitative results of two attention maps with the highest response for the first layer of the FactorVAE (row 2 and 3) and AD-FactorVAE (row 4 and 5) with $\lambda = 40$; $\gamma = 40$ is used for both models. Row 1 shows the ground truth.

Figure 3: Results for latent space disentangling with the FactorVAE and AD-FactorVAE.

For the results corresponding to the reproduction of the third claim about state-of-the-art disentanglement, see figure 3. The left figure shows the quantitative results, this is the reproduction of figure 8 from Liu et al.. The figure on the right shows the reproduction of the qualitative results of the attention maps generated for both a FactorVAE and an AD-FactorVAE model, it corresponds to figure 9 from Liu et al..

### 3.3 Discussion of Results

First of all, for the MNIST experiments, we were able to replicate the results shown in table 4. However, there existed some variation in the quality of the generated anomaly detection maps that was not present in the results of the authors. Image samples resembling the ones used by the authors in table 4 generally show good performance, but for many other image samples, the model did not perform as well. Because the authors mostly show a homogeneous collection of digits with little variation in table 4, it is difficult to determine whether our results are lacking or if the authors just decided to show only the images with the best performance. Our best performing results do, however, closely resemble their results, which substantiates this claim.

Secondly, for the UCSD Ped1 dataset, Liu et al. score a high AUROC score for all layers, where we only score high for one layer. Moreover, their highest score of 0.92 is substantially higher than our score of 0.802. After hyperparameter search, we were able to improve our best results to 0.858 but were unable to match their result. Several explanations are possible: On the one hand, their training setup can differ from ours. They might use another data augmentation pipeline or it is possible that other training hyperparameters are used since it is unclear which optimizer and loss function they use and for how many epochs they train. On the other hand, their calculation of the AUROC score could differ from ours. This can be substantiated by the fact that when we tried to reproduce their baseline of 0.86 by taking the difference between input and the VAE its reconstruction, we obtained a substantially lower baseline of 0.701. We were not able to find out how Liu et al. computed the AUROC score, however, we used the scikit-learn implementation, which uses a standard method. At last, we suggest using a new baseline for evaluating this dataset by taking the difference between the input and the average VAE reconstruction of the latent space, which is essentially an image of an empty pedestrian walkway. Taking this difference resulted in an AUROC score of 0.921, which demonstrates that adding the authors explainable VAE model does not improve anomaly detection for this dataset.

For anomaly detection on the MvTec dataset, most of the reproduced experiments achieved lower scores than the paper. It is noteworthy, however, that the classes leather, capsule and hazelnut all showed decent AUROC scores, similar to the paper or in the case of the capsule even higher results. From these three classes, the significance of the anomaly detection as claim 2 described is at least established. Although the divergent results of the capsule and metal nut class indicate our implementation is different from the one by Liu et al.. Potentially the slight difference in network architecture that resulted from their infeasible network description, in combination with longer training time, could increase the performance to match the state-of-the-art results. Lastly, the results from the capsule class indicate that their implementation is capable of improvements for at least some classes.

Finally, for the latent space disentanglement experiments, the results show that, just as with the results of Liu et al., our reproduction of the AD-FactorVAE with $lambda = 40$ outperforms the standard AD-FactorVAE for the disentanglement metric, without increasing the reconstruction loss. However, it does not do so by as great a margin as claimed in the original paper: we get an improvement of around 0.05, whilst they state an improvement of around 0.09. This could be due to us using a slightly different model, but we found this to be unlikely as we first used a model more similar to theirs but later on switched to the current model. Both setups gave similar results, but ours was slightly faster. Liu et al. also show qualitative results to indicate that the latent space disentanglement is visibly better in the resulting attention maps. We found this to be untrue, as from figure 3b it is not clearly visible that the two latent dimensions are visibly more dissimilar for the AD-FactorVAE than for the FactorVAE. Qualitative analysis is, however, not representative as only a very small part of the dataset was checked, but this is also the case for Liu et al.

# 4 Discussion

## 4.1 Reproducibility

For this reproducibility challenge, there were some difficulties in reproducing the paper by Liu et al., although there were also a few parts that proceeded more smoothly. Reproducing the MNIST experiments, for example, was relatively easy, as the was code available on GitHub repository of the authors. In order to get results that were more consistent with the ones shown in their paper, only minimal hyperparameter tweaking was required, but in general, the experiment could be reproduced by simply running the provided code. Furthermore, implementing the FactorVAE with the dSprites dataset was also not too difficult, as the found implementation was very well structured and documented.

The most difficult part of reproducing the results of the paper arose from the brief and often partial descriptions of the author's implementations. An important example is the ReLU activation that appeared in the paper but was missing in the code. We ran various preliminary experiments comparing the performance of ReLU and absolute, where the absolute operation showed much better performance. For this reason, we suspect that the ReLU in the paper might have been a mistake and the absolute operation is the appropriate function they use. In addition, even though a supplementary document was given, training specifics like the number of epochs or the use of a learning rate scheduler were not mentioned anywhere. Another important detail that remained unclear was which target layers were used for creating the attention maps for different models. At certain points in the paper, for example the UCSD-Ped1 experiments, the authors clearly state which layers were used, but for all other experiments this detail remained unclear or was never mentioned at all. Furthermore, while the supplementary document gave some insight into the scaling and rotating of input images, no mention was given whether the input was normalized. Although it is often considered common practice to normalize input images before training, the lack of mention in the paper makes it difficult to assume anything. In addition, implementing the disentanglement metric proposed by Kim and Mnih [2018] was difficult, as it was both hard to interpret the implementation stated in the paper and the results were not similar to those for the original model at first.

## 4.2 Conclusion

From the results, we can conclude that the first claim Liu et al. make, that they can generate visual attention maps for VAEs, is true. We were successfully able to generate these attention maps and could apply them to anomaly detection as well. For the MNIST experiments, this too led to results that were similar to the paper. However, for the UCSD-Ped1 experiments, the explainable VAE model of the authors actually performed worse than our own baseline. Moreover, we were not able to support the authors their claim that they achieve state-of-the-art on the MVTec dataset. It is worth noting that we may have arrived at these different results due to the unclarity of the authors their implementation details. Therefore, we conclude that their second claim partially holds, as our experiments show that it is possible to use their attention maps for anomaly detection, but we cannot always match their performance. Finally, the third claim, that attention can help with latent space disentanglement, also holds. Although our found results are not as good as claimed by Liu et al., they are better than those for the set baseline that does not use the attention maps.

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

 # A  Model Architectures

| Network | Layer | Output Dimensions |
|---|---|---|
| Encoder | Conv 2D, $4 \times 4$, 64,2,1 | $14 \times 14 \times 64$ |
| | ReLU | $14 \times 14 \times 64$ |
| | Conv 2D, $4 \times 4$, 128,2,1 | $7 \times 7 \times 128$ |
| | ReLU | $7 \times 7 \times 128$ |
| | Flatten | 6272 |
| | Linear | 1024 |
| | ReLU | 1024 |
| | Linear | 32 |
| Decoder | Linear | 1024 |
| | ReLU | 1024 |
| | Linear | 6272 |
| | ReLU | 6272 |
| | Unflatten | $7 \times 7 \times 128$ |
| | ReLU | $7 \times 7 \times 128$ |
| | ConvTr 2D, $4 \times 4$, 64,2,1 | $14 \times 14 \times 64$ |
| | ReLU | $14 \times 14 \times 64$ |
| | ConvTr 2D, $4 \times 4$, 1,2,1 | $28 \times 28 \times 1$ |
| | Sigmoid | $28 \times 28 \times 1$ |

Figure 4: One-class Vanilla VAE for MNIST.

| Network | Layer | Output Dimensions |
|---|---|---|
| Encoder | Conv 2D, $4 \times 4$, 64,2,1 | $50 \times 50 \times 64$ |
| | ReLU | $50 \times 50 \times 64$ |
| | Conv 2D, $4 \times 4$, 128,2,1 | $25 \times 25 \times 128$ |
| | ReLU | $25 \times 25 \times 128$ |
| | Conv 2D, $4 \times 4$, 256,2,1 | $12 \times 12 \times 256$ |
| | ReLU | $12 \times 12 \times 256$ |
| | Flatten | 36864 |
| | Linear | 1024 |
| | ReLU | 1024 |
| | Linear | 32 |
| Decoder | Linear | 1024 |
| | ReLU | 1024 |
| | Linear | 36864 |
| | ReLU | 36864 |
| | Unflatten | $256 \times 12 \times 12$ |
| | ReLU | $256 \times 12 \times 12$ |
| | ConvTr 2D, $5 \times 5$, 128,2,1 | $25 \times 25 \times 128$ |
| | ReLU | $25 \times 25 \times 128$ |
| | ConvTr 2D, $4 \times 4$, 64,2,1 | $50 \times 50 \times 64$ |
| | ReLU | $50 \times 50 \times 64$ |
| | ConvTr 2D, $4 \times 4$, 1,2,1 | $100 \times 100 \times 1$ |
| | Sigmoid | $100 \times 100 \times 1$ |

Figure 5: One-class Vanilla VAE used by authors for UCSD-Ped1.

| Network | Layer | Output Dimensions |
|---|---|---|
| Encoder | Resnet18 | 32 |
| Decoder | Linear | 1024 |
| | Linear | $1024 \times 4 \times 4$ |
| | ConvTr 2D, $4 \times 4$, 512,2,1 | $8 \times 8 \times 512$ |
| | BatchNorm | $8 \times 8 \times 512$ |
| | ReLU | $8 \times 8 \times 512$ |
| | ConvTr 2D, $4 \times 4$, 256,2,1 | $16 \times 16 \times 256$ |
| | BatchNorm | $16 \times 16 \times 256$ |
| | ReLU | $16 \times 16 \times 256$ |
| | ConvTr 2D, $4 \times 4$, 128,2,1 | $32 \times 32 \times 128$ |
| | BatchNorm | $32 \times 32 \times 128$ |
| | ReLU | $32 \times 32 \times 128$ |
| | ConvTr 2D, $4 \times 4$, 64,2,1 | $64 \times 64 \times 64$ |
| | BatchNorm | $64 \times 64 \times 64$ |
| | ReLU | $64 \times 64 \times 64$ |
| | ConvTr 2D, $4 \times 4$, 32,2,1 | $128 \times 128 \times 32$ |
| | BatchNorm | $128 \times 128 \times 32$ |
| | ReLU | $128 \times 128 \times 32$ |
| | ConvTr 2D, $4 \times 4$, 3,2,1 | $256 \times 256 \times 3$ |
| | Sigmoid | $256 \times 256 \times 3$ |

Figure 6: Resnet18 VAE we used.

| Network | Layer | Output Dimensions |
|---|---|---|
| Encoder | Input Image | $64 \times 64$ |
| | Conv 2D, $4 \times 4$, 32,2,1 | $32 \times 32 \times 32$ |
| | ReLU | $32 \times 32 \times 32$ |
| | Conv 2D, $4 \times 4$, 32,2,1 | $16 \times 16 \times 32$ |
| | ReLU | $16 \times 16 \times 32$ |
| | Conv 2D, $4 \times 4$, 64,2,1 | $8 \times 8 \times 64$ |
| | ReLU | $8 \times 8 \times 64$ |
| | Conv 2D, $4 \times 4$, 64,2,1 | $4 \times 4 \times 64$ |
| | ReLU | $4 \times 4 \times 64$ |
| | Conv 2D, $4 \times 4$, 128,1,1 | $1 \times 1 \times 128$ |
| | ReLU | $1 \times 1 \times 128$ |
| | Conv 2D, $1 \times 1$, 32,1,0 | 32 |
| | Conv 2D, $1 \times 1$, 32,1,0 | 32 |
| Decoder | Input | $\mathbb{R}^{32}$ |
| | Conv 2D, $1 \times 1$, 128,1,0 | 128 |
| | ReLU | $1 \times 1 \times 128$ |
| | ConvTr 2D, $4 \times 4$, 64,1,0 | $4 \times 4 \times 64$ |
| | ReLU | $4 \times 4 \times 64$ |
| | ConvTr 2D, $4 \times 4$, 64,2,1 | $8 \times 8 \times 64$ |
| | ReLU | $8 \times 8 \times 64$ |
| | ConvTr 2D, $4 \times 4$, 32,2,1 | $16 \times 16 \times 32$ |
| | ReLU | $16 \times 16 \times 32$ |
| | ConvTr 2D, $4 \times 4$, 32,2,1 | $32 \times 32 \times 32$ |
| | ReLU | $32 \times 32 \times 32$ |
| | ConvTr 2D, $4 \times 4$, 1,2,1 | $64 \times 64 \times 1$ |

Figure 7: FactorVAE used by authors.

 **B Qualitative Results on MVTec Dataset**

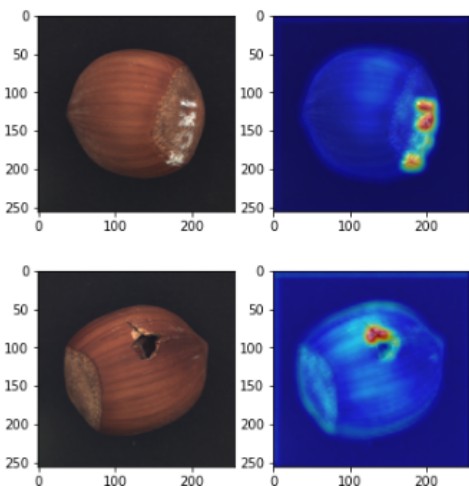

Figure 8: Reproduced images of the attentionmap on the hazelnut class

