# OpenReview forum: "Reproducibility Report: Towards Visually Explaining Variational Autoencoders"
_ML_Reproducibility_Challenge/2020 — Reject_

### Official Review · AnonReviewer2 · 2021-02-27
**Review of the reproducibility report of "Towards visually explaining variational autoencoders"**

**Rating:** 4
**Confidence:** 3

**Review:**

## Overview
This report aims to investigate the reproducibility of "Towards visually explaining variational autoencoders". The reproducibility summary is insightful and details the step necessary as well as the difficulty they encountered while trying to reproduce the results of the paper. It shows that even if more and more authors are now making code available through different means (paper with code, GitHub), there are still some ways to go before reproducibility in AI is the norm.

## Quality:
The quality of the report is mixed.
### Pros
- The methodology is well explained and the plan to reproduce the results is solid and well thought out. The authors did an extensive effort to reproduce and also re-implement the results from the original paper. The code they made available online to reproduce the results from the report work well, they are sufficiently commented and I could reproduce the results from the report. Contrary to the original paper where a large proportion of the code was missing.
- The dataset, parameters, and setup for the experiments are well presented.
- The results are well presented and show that the implementation is functional.

### Cons
However, the paper is not well structured in my opinion:
- the last paragraph of the introduction should absolutely state the results of the report upfront. It should be written in the introduction what was achieved by the paper, the reader should not have to read the whole paper to have the most relevant information. The report presents the three major claims that are made by the original paper and should answer them in the same paragraph.
- the methodology section is missing a high-level explanation of the concepts they are aiming to implement. As it stands, a reader not familiar with the original paper could not follow the experiments that are being presented. For example, the disentanglement problem is not defined nor explained in the report.
- section 4.1 "Discussion of results" should have been included in the section results. This makes the results and the figure hard to follow.

## Clarity
The paper is well written and the article is easy to read. However, it would need more proofreading.
- What are the equation referring to at line 87 and line 130 ??.
- Why in table 1 (a) line "conv 2" Our reproduction "0.644", our best "0.320" the best results is lower than the reproduction?
Also, the references are inconsistent, for example, citation lines 320-322 and citation lines 339-341 are citing the same conference and are not formatted in the same way.

## Final words
The authors of the report did an excellent work reproducing the results from the original paper and had an interesting take on why their results are not consistent with the original paper, however, the paper is poorly presented and is missing crucial high-level information on the main concept they are trying to present. Finally, the review is supposed to be double blind, however, the GitHub repository where the authors made the code available was not anonymized.

**Familiar With The Original Paper:**

I have read the original paper

**Reproducibility Summary:**

Report has summary

---

### Official Review · AnonReviewer3 · 2021-03-01
**Detailed report, A successful replication**

**Rating:** 8
**Confidence:** 4

**Review:**

The report clearly summarizes the problem statement of the original paper "Towards Visually Explaining Variational Autoencoders".

The submission has covered all experiments in the original paper. Particularly, when the original implementations are missing, the authors of the submission use good reasonings and additional material to decide their own setting.

The authors also use good reasonings to discuss why their results are different from the original paper, especially on the difference for the UCSD Ped1 dataset

The submission itself has covered essential implementation details. After reading this submission, I don't see any further details that are needed for the implementation.

Minor comments:
1) It would be helpful to provide a learning rate scheduler in supplementary.
2) Do the authors try to choose two latent dimensions not randomly, but through a search of which two dimensions are the most informative (either for the overall entropy, or the mutual information between the input and the latent encodings?). Is it possible/feasible to compute these metrics without exhaust the GPU?


**Familiar With The Original Paper:**

I have read the original paper

**Reproducibility Summary:**

Report has summary

---

### Official Review · AnonReviewer1 · 2021-03-11
**Good description of a failure to reproduce results with good analysis and suggestions**

**Rating:** 8
**Confidence:** 3

**Review:**

Reproducibility Summary: Included, covers the relevant topics

Scope of Reproducibility: Re-used existing code for MNIT experiments; custom implementation for the other datasets

Code: linked to github repo

Communication with Original Authors: the authors were contacted, but no response was received directly. It seems like a reasonable and fair effort was made.

Hyper-parameter Search: The reproduction largely relies on its own implementation

Ablation Study:

Discussion: Clear discussion of ways that the original paper is hard to reproduce. Including hard to follow code and mismatches between the implementation and the description in the paper.

Recommendations for Reproducibility: Well described in section 4.2

Results Beyond Paper: New baseline proposed and experimented with

Overall Organization/Clarity: well written paper, with a few minor errors in language.

Errata

L.87 equation reference is broken

**Familiar With The Original Paper:**

I have not read the original paper

**Reproducibility Summary:**

Report has summary

---

### Decision · Program_Chairs · 2021-03-31

**Decision:**

Reject

**Comment:**

Overall reviews and/or the paper content not good enough for the AC to recommend to the journal.